# Metabolomic-Based Comparison of Traditional and Industrial *Doenjang* Samples with Antioxidative Activities

**DOI:** 10.3390/foods10061377

**Published:** 2021-06-15

**Authors:** Song-Hui Soung, Sunmin Lee, Seung-Hwa Lee, Hae-Jin Kim, Na-Rae Lee, Choong-Hwan Lee

**Affiliations:** 1Department of Bioscience and Biotechnology, Konkuk University, Seoul 05029, Korea; songhee2706@naver.com (S.-H.S.); duly123@naver.com (S.L.); michelle3690@gmail.com (N.-R.L.); 2Experiment Research Institute, National Agricultural Products Quality Management Service, Gimcheon si 39660, Korea; shlee96@korea.kr (S.-H.L.); asarela00@korea.kr (H.-J.K.)

**Keywords:** *doenjang*, non-targeted metabolite profiling, biochemical phenotypes, correlation analysis

## Abstract

Numerous varieties of *doenjang* are manufactured by many food companies using different ingredients and fermentation processes, and thus, the qualities such as taste and flavor are very different. Therefore, in this study, we compared many products, specifically, 19 traditional *doenjang* (TD) and 17 industrial *doenjang* (ID). Subsequently, we performed non-targeted metabolite profiling, and multivariate statistical analysis to discover distinct metabolites in two types of *doenjang*. Amino acids, organic acids, isoflavone aglycones, non-DDMP (2,3-dihydro-2,5-dihydroxy-6-methyl-4H-pyran-4- one) soyasaponins, hydroxyisoflavones, and biogenic amines were relatively abundant in TD. On the contrary, contents of dipeptides, lysophospholipids, isoflavone glucosides and DDMP-conjugated soyasaponin, precursors of the above-mentioned metabolites, were comparatively higher in ID. We also observed relatively higher antioxidant, protease, and *β*-glucosidase activities in TD. Our results may provide valuable information on *doenjang* to consumers and manufacturers, which can be used while selecting and developing new products.

## 1. Introduction

*Doenjang,* fermented soybean paste, has played an important role as a food ingredient and protein source in Korea for hundreds of years. For a long time, various types of *doenjang* have been developed and generally used in many dishes to enhance taste [1]. Many studies revealed that the abundance of certain amino acids in *doenjang* influences flavor and taste (e.g., umami, sweetness and bitterness) [2]. Furthermore, numerous health-beneficial effects (e.g., anti-cancer, antioxidant and antimetastatic) of *doenjang* have been studied [3,4,5,6,7,8].

*Doenjang* is made from *meju*, which is a fermented soybean lump. To make *meju*, soybeans are pretreated sequentially by soaking, steaming, crushing and molding into solid blocks [9]. The *meju* is brined and fermented for several months, and then, separated into liquid and solid parts for further fermentation to produce soy sauce and *doenjang* (soy paste), respectively. The solid component is fermented for a few more months to years for maturing of the *doenjang* [10]. The fermentation leads to the enhancement of many nutritional metabolites (e.g., amino acids, fatty acids, organic acids, minerals and vitamins) [11] and bioactive metabolites (e.g., isoflavonoids and soyasaponins) in the *doenjang* through the degradation of ingredients. The metabolite contents in a variety of products are significantly distinguished due to many factors such as different fermentation microorganisms, fermentation periods, and ingredients [1].

Those factors are mainly manipulated to improve the quality of *doenjang.* Generally, traditional *doenjang* (TD) is characterized by varied fermentation periods and diverse fermentation microorganisms, which are inoculated from rice straw [9,11]. Therefore, the TD sometimes shows inconsistent qualities such as metabolite contents in different batches. On the other hand, industrial *doenjang* (ID) is manufactured by inoculating few microorganisms and fermented under a controlled environment for a certain period [12]. Thus, ID is easier to reproduce and maintain the quality than TD. However, it is known that ID lacks deep and rich flavors and tastes as compared to TD, which might be caused by a limited number of fermentation microorganisms [13].

Along with this, numerous studies have been conducted to elucidate different properties of various *doenjang* such as sensory properties [11], physiochemical quality [12], and metabolites [14,15]. Most of the metabolome studies considered only a handful of *doenjang* products, which cannot provide general information of *doenjang*. In addition, it is difficult to understand the different traits of various *doenjang* comprehensively because the analyzed metabolites were limited to amino acids, flavonoids, and biogenic amines [16]. Therefore, in current study, we purchased 36 different commercially available *doenjang* to compare metabolite contents and evaluate physicochemical properties. We categorized the purchased products into two groups, traditional *doenjang* (TD) and industrial *doenjang* (ID), according to the manufacturing process used. Next, we performed an untargeted metabolite profiling using the GC-TOF-MS and UHPLC-LTQ-Orbitrap-ESI-MS/MS. Additionally, different metabolite contents in the two groups were depicted in a metabolic pathway map. Finally, we evaluated the correlation of metabolite contents with antioxidant activities and enzymatic activities.

## 2. Materials and Methods

### 2.1. Chemicals and Reagents

HPLC-grade water, methanol, and acetonitrile were purchased from Fisher Scientific (Pittsburgh, PA, USA). Diethylene glycol was obtained from Junsei Chemical Co., Ltd. (Tokyo, Japan). Formic acid, pyridine, methoxyamine hydrochloride, N-methyl-N-(trimethylsilyl) trifluoroacetamide (MSTFA), potassium persulfate, 2,2′-azino-bis (3-ethylbenzothiazoline-6-sulfonic acid) diammonium salt (ABTS), 6-hydroxy-2,5,7,8-tetramethylchromane-2-carboxylic acid (Trolox), sodium acetate, acetic acid, 2,4,6-Tris (2-pyridyl)-s-triazine (TPTZ), hydrochloric acid (HCl), iron (III) chloride hexahydrate, sodium carbonate, Folin–Ciocalteu’s phenol, sodium hydroxide, gallic acid, and naringin were obtained from Sigma-Aldrich (St. Louis, MO, USA).

### 2.2. Sample Information and Preparation

Thirty-six *doenjang* products, including 19 kinds of TD and 17 kinds of ID, were obtained from various markets. Detailed information of the purchased *doenjang* (e.g., manufacturer, fermentation period and ingredients) is represented in Appendix A. After purchasing, the samples were stored at 4 °C until analysis. All *doenjang* samples were lyophilized for 3 days and powdered. One mL of 80 % aqueous methanol was added to each sample (100 mg) in a 2 mL eppendorf tube, and then, homogenized using a Retsch MM400 Mixer Mill (Retsch GmbH & Co, Haan, Germany) at 30 s^−1^ for 5 min. The resulting mixtures were centrifuged at 13,000× *g* for 10 min at 4 °C. After centrifugation, the solvent was separated, and the same procedure was repeated twice. Subsequently, the sample extracts were filtered using 0.22-µm polytetrafluoroethylene (PTFE) filters (Chromdisc, Daegu, Korea), and evaporated using a speed vacuum concentrator (Biotron, Seoul, Korea). The dried extracts were dissolved in 80% methanol to adjust the final concentration to 10,000 ppm, and then, filtered again for further instrumental analyses.

### 2.3. GC-TOF-MS Analysis

For GC-TOF-MS analysis, the redissolved samples were dried using a speed vacuum concentrator. The dried samples were derivatized as follows: (1) the oximation was performed by adding 50 µL of 20 mg/mL methoxyamine hydrochloride in pyridine to the dried samples and incubating at 30 °C for 90 min (2) the silylation was performed by mixing 50 µL of MSTFA to the mixture and incubating at 37 °C for 30 min. The derivatized samples were filtered using Millex GP 0.22 µm filters (Merck Millipore, Billerica, MA, USA).

GC-TOF-MS analysis was performed using an Agilent 7890A GC system (Santa Clara, CA, USA) equipped with Rtx-5MS (length 30 m, inner diameter 0.25 mm, J&W Scientific, Folsom, CA, USA) coupled with an Agilent 7693 autosampler and a Pegasus^®^ HT TOF-MS (LECO Corp., St. Joseph, MI, USA). The analysis parameters were adapted from a previous study [2]. Three biological replications were analyzed for each sample.

### 2.4. UHPLC-LTQ-Orbitrap-ESI-MS/MS Analysis

UHPLC-LTQ-Orbitrap-ESI-MS/MS analysis was performed using a UHPLC system equipped with a Vanquish binary pump H system (Thermo Fisher Scientific, Waltham, MA, USA) coupled with auto-sampler and column compartment. A Phenomenex KINETEX^®^ C18 column (100 mm × 2.1 mm, 1.7 µm particle size; Torrance, CA, USA) was used to separate a sample analyte. The analytical parameters were adopted from a study reported by Kwon et al. [17].

### 2.5. Data Processing and Multivariate Statistical Analysis

The raw data of GC−TOF−MS and UHPLC-LTQ-Orbitrap-ESI-MS/MS were converted to NetCDF (*.cdf). Subsequently, the NetCDF (*.cdf) files were processed with the MetAlign software (http://www.metalign.nl (accessed on 21 May 2021) Wageningen, Netherlands) for data alignment in accordance with peak mass (m/z), and retention time (min) [18]. Multivariate statistical analysis was conducted using SIMCA-P+ software (version 15.0.2, Umetrics, Umea, Sweden), and principal component analysis (PCA) and partial least squares discriminant analysis (PLS-DA) modeling were performed to compare the different metabolites among the samples. The discriminant metabolites were selected based on variable importance in the projection (VIP) value >0.7 and *p*-value < 0.05. The selected metabolites were tentatively identified by comparison with various analysis data such as molecular weights, formula, retention time, mass fragment patterns, and mass spectrum of standard compounds, the published references, the chemical dictionary version 7.2 (Chapman and Hall/CRC), an in-house library (off-line database in laboratory made by analyzing standards), commercial databases, such as National Institutes of Standards and Technology (NIST) Library (version 2.0, 2011, FairCom, Gaithersburg, MD, USA), and the Human Metabolome Database (HMDB; http://www.hmdb.ca/ (accessed on 21 May 2021)).

### 2.6. Quantification of Biogenic Amines

In order to quantify biogenic amines in *doenjang*, the amounts of detected biogenic amines (agmatine, histamine, phenylethylamine, putrescine, spermidine, spermine, tryptamine, and tyramine) were calculated using standard curves. Each stock solution was prepared at 1 mg/mL. The standard curves of each target metabolite were ranged from 5 ppm to 500 ppm. Quantification of biogenic amines excluding putrescine were conducted by using UHPLC-Orbitrap-MS/MS. The putrescine was measured by GC-TOF-MS.

### 2.7. Antioxidant Activities (ABTS and FRAP) and TPC Analysis

The antioxidant activities were performed using ABTS and FRAP assays, following the methods reported by Lee et al. [10].

For the TPC assay, the Folin–Ciocalteu colorimetric method was used. Briefly, 0.2 N Folin–Ciocalteu’s phenol reagent (100 µL) was added to 20 µL of each sample in a 96-well plate and incubated at room temperature for 6 min in the dark. Subsequently, 80 µL of 7.5% sodium carbonate solution was supplemented to the mixture, and reacted for 60 min at ambient temperature. The absorbance was determined at 750 nm. The result was presented as the gallic acid equivalent (GAE) concentration (ppm), ranging from 3.91 ppm to 500 ppm using a standard curve. All experiments were performed in triplicate.

### 2.8. Enzyme Activity Assay

Each *doenjang* sample (5 g) was extracted with 45 mL of distilled water by shaking in an orbital shaker at 120 rpm for 1 h. The mixture was centrifuged at 15,000 rpm for 10 min at 4 °C and the supernatant was separated. The supernatant was filtered using a 0.2 μm PTFE filter.

To evaluate protease activity, the extracted *doenjang* sample (1 mL) was mixed with 5 mL of 0.6% casein solution and incubated at 37 °C for 10 min. After incubation, the 5 mL of 0.4 M trichloroacetic acid was supplemented and kept at 37 °C for 30 min to stop the reaction. The reacted mixture was filtered again. The filtrated sample (2 mL), 0.4 M sodium carbonate (5 mL), and 2 N folin reagent (1 mL) were mixed and incubated at 37 °C for 30 min. Next, the reaction mixture was measured at 660 nm using a spectrophotometer (Spectronic Genesys 6, Thermo Electron, Madison, WI, USA).

The *β*-glucosidase activity of *doenjang* samples was determined using the p-nitrophenyl *β*-D-glucopyranoside (pNPG). One mL of extracted *doenjang* sample was mixed with 1 mL of pNPG, and 8 mL of sodium acetate buffer, and then, reacted at 37 °C for 30 min. The enzymatic reaction was ceased by adding 5 mL of 0.4 M sodium carbonate. The absorbance was monitored at 400 nm using a spectrophotometer.

## 3. Results and Discussion

### 3.1. Multivariate Statistical Analysis of Traditional and Industrial Doenjang

In order to evaluate differences of TD and ID, a metabolite profiling approach was harnessed using GC-TOF-MS and UHPLC-LTQ-Orbitrap-ESI-MS/MS. Subsequently, we conducted multivariate analysis based on metabolome analysis results. We used numerous commercially available products, including 19 TD and 17 ID products, to avoid sampling bias. Notably, the PCA and PLS-DA score plot derived from GC-TOF-MS data represents that the metabolite profiles of TD and ID were clearly separated by PC1 (23.0%) and PLS1 (22.8%) (Figure 1A,C). Intriguingly, this shows an obvious pattern in which different products from the same company were clustered in the PCA and PLS-DA plots (Figure 1A,C). For example, a cluster of certain IDs (ID 4, 5, 6 and 7), manufactured by same company, was apparently distinguished from other ID products. Similarly, it showed a cluster in TD as well with TD17 and TD18, which are manufactured from the same producer.

We observed that the PCA and PLS-DA score plots of UHPLC-LTQ-Orbitrap-ESI-MS/MS (based on negative mode) analysis are similar to the GC-TOF-MS analysis score plots (Figure 1B,D). The TD cluster was obviously distinguished from ID by PC1 (10.6%) and PLS1 (10.5%). Similar to the primary metabolites analysis result, the products from the same manufacturing company were clustered based on secondary metabolite profiling results (Figure 1B,D).

Based on the PCA using primary and secondary metabolite analyses, we observed the significantly distinguished pattern between ID and TD. We speculate that the obviously distinct patterns might have occurred due to different factors of fermentation processes such as (1) ingredients, (2) fermentation microorganisms, and (3) fermentation periods [15]. In the current study, we figured out that ingredients of ID are distinctive from TD, which contains wheat or rice. The carbohydrates might be added during the industrial manufacturing of *koji* (mold for food fermentation). Additionally, we hypothesized that the distinct pattern of two types of *doenjang* (TD vs. ID) occurred as a result of the different microbial communities. Generally, TD is inoculated naturally by undefined microorganisms, existing in the air and rice straw, whereas ID is fermented by defined microorganisms which were inoculated at the beginning of fermentation. Due to the different microbial consortiums in various *doenjang*, the resulting metabolite contents in *doenjang* might be substantially different. From that, we can guess that the unique metabolism of certain microorganisms in the microbial community affected the metabolite contents during fermentation [9]. Additionally, different metabolite profiles of various types of *doenjang* might be affected by dynamic changes of the bacterial and fungal communities during fermentation [19]. Most ID are manufactured within few months under controlled fermentation systems, while the TD fermentation periods are comparatively longer, at least 6 months to a few years. We expect that the distinct patterns of ID and TD in score plots were also influenced by the fermentation period, which can be associated with the final metabolite contents.

### 3.2. Primary Metabolite Profiling of Traditional and Industrial Doenjang

In order to investigate the different metabolites in the TD and ID groups, we selected significantly discriminated metabolites based on the VIP value (>0.7) using PLS-DA and *p*-value < 0.05 (Figure 1C). As a result, a total of 19 primary metabolites, including 3 organic acids, 13 amino acids, 2 sugar and sugar derivative, and putrescine, were tentatively identified from GC-TOF-MS data analysis (Appendix A). The relative contents of significantly different metabolites in TD and ID were depicted in the metabolic pathway (Figure 2); these were calculated based on the peak area of each chromatogram. Intriguingly, organic acids were observed to be relatively higher in TD than ID. The organic acids in *doenjang* determine flavors such as acidity and sourness, which are major considerations when purchasing fermented foods [7,20]. We can speculate that the relatively abundant organic acids in TD might be achieved by anonymous fermentation microbes during long fermentation periods [7,16]. In addition, carbohydrates such as glucose and mannose, except for myo-inositol, were comparatively higher in ID than TD, which might have resulted from the addition of cereals (wheat or rice) into ID while making *meju*. We can guess that the carbohydrates in ID were produced from cereals via α-amylase by fermentation microorganisms, corresponding with a previous study in which wheat *koji* has high α-amylase activity [8,14]. Interestingly, our results showed that most amino acids excluding glutamic acid were significantly abundant in TD, which might be affected by a long fermentation period [3]. Moreover, this result is consistent with relatively higher protease activities in TD, which are secreted from microorganisms (Figure 3E) [6]. It is known that the amino acids in food are not only a good source of nutrients, but also one of the important factors which contribute to flavor. For example, some amino acids such as alanine, serine, proline, and threonine are relevant to the sweetness of foods [2], whereas glutamic acid and aspartic acid are the main contributors to umami taste [21]. According to our primary metabolite profiling result, TD is a more attractive product to consumers because it contains relatively abundant metabolites of sweetness and umami taste, which are preferable tastes when selecting *doenjang* [12].

### 3.3. Secondary Metabolite Profiling and Metabolic Pathway Analysis of Traditional and Industrial Doenjang

A total of 28 discriminant secondary metabolites, including 6 isoflavonoids, 5 soyasaponins, 6 lysophospholipids, 2 amino acids, 2 dipeptides and 7 biogenic amines, were tentatively identified from UHPLC-Orbitrap-MS/MS data analysis (Appendix A). In order to compare secondary metabolites in the TD and ID groups, we identified significantly different metabolites based on the VIP value (>0.7) using PLS-DA and *p*-value of < 0.05 (Figure 1D).

The relative levels of lysophospholipid in ID such as lyso PE (18:2), lyso PE (16:0), lyso PE (18:1), lyso PC (18:2), lyso PC (15:0), and lyso PC (18:1) were higher than TD. In a previous study, it was described that the contents of phospholipid are negatively correlated with the fermentation period since it degraded by lipase or phospholipase during fermentation [22].

Flavonoids, including isoflavone aglycones and hydroxyisoflavones, which are well-known antioxidants, were observed in both TD and ID. Notably, ID contained relatively higher contents of isoflavone glucoside, while TD showed a comparative abundance of hydroxyisoflavones and isoflavone aglycones. It is reported that isoflavone glucosides in soybeans are converted to their corresponding aglycone forms via *β*-glucosidase [10]. Based on our metabolite profiling result, we assumed that the relatively abundant isoflavone aglycones were produced from glucoside forms through *β*-glucosidase, which is concurrent with the enzyme activity assay results (Figure 3F). Relatively abundant hydroxyisoflavones in TD, including hydroxygenistein, hydroxydaidzein, and hydroxyglycitein, might be produced by *A. oryzae* according to a previous study [23]. Based on the above, we speculated that the influence of *A. oryzae* on *doenjang* fermentation was relatively higher in TD than ID.

In current study, the DDMP-conjugated soyasaponin βa, a precursor of non-DDMP forms, was higher in ID, whereas the non-DDMP-conjugated soyasaponin A2, I, II, and IV, derived from the βa form, were comparatively higher in TD. Consistently, it is reported that the DDMP-conjugated soyasaponins are mainly found in raw soybeans, whereas non-DDMP-conjugated soyasaponins (hydrolyzed form) are mainly found in processed soy products [24,25,26].

We observed biogenic amines, generally found in a variety of foods (e.g., beverages, and fermented foods), which can be used as a potential food freshness or spoilage indicator [27]. Tyramine, 2-phenylethylamine, tryptamine, histamine, agmatine, spermine, and spermidine were detected in both TD and ID. Additionally, we performed the quantification of biogenic amines in *doenjang* samples because the efficacy of biogenic amines for humans is substantially different depending on the concentration. For example, these are harmful at high concentrations, causing food poisoning or cancer [16]. However, low levels of biogenic amines in food do not cause risks to human health since the amines are detoxified quickly in human intestines [28]. Furthermore, the biogenic amines are helpful to the human central nervous system by functioning as neurotransmitters at low concentrations [16]. The European Food Safety Authority suggested that foods containing less than 1000 mg/kg for total biogenic amine are safe for consumption [29]. Moreover, guidance levels of each biogenic amine such as β-phenylethylamine, histamine, and tyramines are 30 mg/kg, 100 mg/kg, and 100–800 mg/kg, respectively [30,31]. We summarized the amounts of each biogenic amine in both TD and ID (Appendix A). The total biogenic amines in all *doenjang* samples ranged between 0.0009 and 0.0046 mg/kg, which is considered to be a safe and health-beneficial level. The types and amounts of biogenic amines in fermented food might be affected by the environmental conditions (e.g., microbial flora, and fermentation process) [32]. In our results, the levels of most biogenic amines excluding spermine and spermidine were comparatively higher in TD than ID. Interestingly, the precursors of detected biogenic amines, such as histidine, tryptophan, tyrosine, phenylalanine, ornithine, and arginine, were also higher in TD. A part of free amino acids might be converted to the biogenic amines via decarboxylase, which is secreted by fermentation microorganisms. The optimal pH of the decarboxylase is between 3.0 and 6.0 [33,34]. In our results, we observed an insignificant difference in pH between TD and ID (pH 5.0–5.5) that can provide optimal conditions to produce the biogenic amines (Figure 3D). Furthermore, it was reported that *doenjang* fermented by defined microorganisms has a lower chance of producing biogenic amines with a controlled fermentation system, but anonymous microorganisms by natural fermentation lead the *doenjang* to produce comparatively higher contents of biogenic amines [35]. Overall, we assumed that the differences of secondary metabolite contents between two types of *doenjang* were affected by the microbial community and the contents of precursors.

### 3.4. Correlation Analysis between Metabolome and Biochemical Phenotypes

In order to obtain information regarding the bioactive metabolites in *doenjang*, we performed antioxidant activity assays (ABTS, FRAP, and TPC) and enzyme activity assays (protease and *β*-glucosidase). After that, we evaluated the statistical correlation between antioxidant activities, protease activities and metabolites in TD and ID using Pearson’s correlation analysis (Figure 4).

As expected, TD showed significantly higher ABTS, FRAP, and TPC activities (*p*-value < 0.05) than ID (Figure 3A–C). The ABTS, FRAP, and TPC activities showed positive correlation with some amino acids, isoflavone aglycones, hydroxyisoflavones, and non-DDMP-conjugated soyasaponins (Figure 4). It was reported that the positively correlated amino acids, including tryptophan, methionine, histidine, tyrosine, and cysteine, have free radical scavenging potentials [36]. Even though the antioxidant activities of amino acids are relatively lower than other well-known antioxidant metabolites, it seems that the amino acids partially contribute to antioxidant ability of *doenjang* [36]. Additionally, the isoflavone aglycone, hydroxyisoflavones, and non-DDMP-conjugated soyasaponins showed strong correlation with antioxidant function, as described elsewhere [23,37].

Most amino acids and dipeptides showed significant positive correlation with protease activity. As we described above, relatively higher contents of amino acids in TD were observed, which is consistent with higher protease activity. In addition, they showed significant positive correlation between *β*-glucosidase and its relevant metabolites such as isoflavone aglycones, hydroxyisoflavones, and non-DDMP-conjugated soyasaponins in TD. Through that, we assumed that the biofunctional activities of *doenjang* are highly influenced by enzymes from microbial flora which can be different depending on the fermentation process.

## 4. Conclusions

*Doenjang* differs extensively depending on the manufacturing process used and such variables as the fermentation microorganisms, fermentation period, and ingredients. However, the metabolomic differences of various types of *doenjang* have not been revealed so far. Therefore, we performed non-targeted metabolite profiling of commercially available TD and ID products. We observed the significant separation of TD and ID in both primary and secondary metabolites. Generally, it is well-known that primary metabolites influence taste and nutrition. On the other hand, the secondary metabolites affect the functional properties of foods. In our study, the TD contained comparatively higher contents of nutritional compounds such as amino acids and organic acids. Isoflavone aglycones and non-DDMP soyasaponins, which contribute to the bioactivity of *doenjang*, were abundant in TD, while the DDMP-conjugated soyasaponins, isoflavone glucoside and lysophospholipids were observed abundantly in ID, which can be fermented further to some metabolites in TD. In addition, the antioxidant activities, based on ABTS, FRAP and TPC assays, and protease and β-glucosidase activities, were higher in TD. The metabolite contents in TD and ID that are relevant to functional properties were significantly different, which might be affected by ingredients, microbes, fermentation period and storage period. The current study, which compared metabolites and biochemical properties using 36 different *doenjang* products, can provide valuable information to both consumers and manufacturers. In the future, time-resolved metabolome analysis along with microbial community analysis should be studied to understand the dynamic changes in metabolites and microorganisms in *doenjang* during fermentation.

## Figures and Tables

**Figure 1 foods-10-01377-f001:**
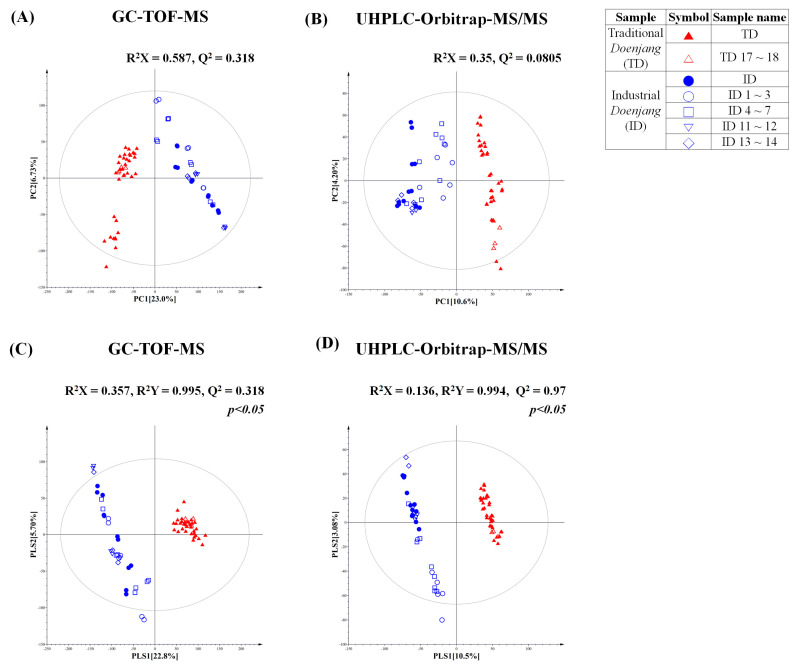
PCA and PLS-DA score plots of traditional and industrial *doenjang* (▲ (TD): traditional *doenjang*, ● (ID): industrial *doenjang*) based on GC-TOF-MS (**A**,**C**) and UHPLC-Orbitrap-MS/MS (**B**,**D**). Unfilled symbols represent samples from the same manufacturers.

**Figure 2 foods-10-01377-f002:**
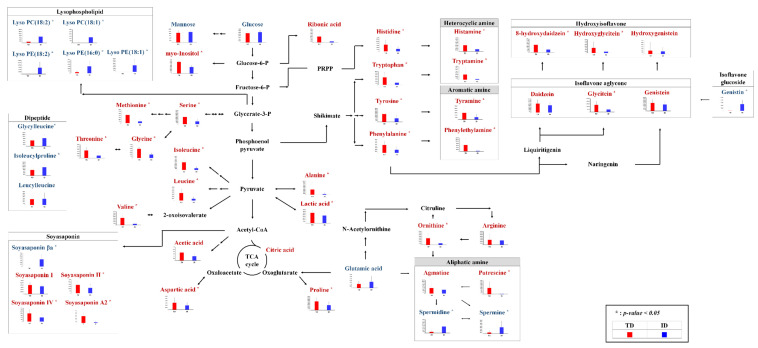
Scheme of the primary and secondary metabolic pathway and relative metabolite contents of traditional and industrial *doenjang*. The metabolic pathway was adopted from the KEGG database. The Y-axis of the graph represents peak areas of respective metabolites. Color of metabolite names represents comparatively abundant metabolites in traditional (red) and industrial (blue) *doenjang*.

**Figure 3 foods-10-01377-f003:**
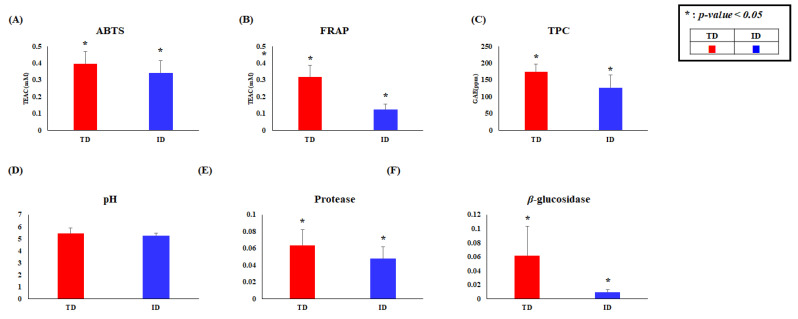
Antioxidant and enzyme activity analyses. (**A**) ABTS, (**B**) FRAP, (**C**) TPC, (**D**) pH, (**E**) protease and (**F**) *β*-glucosidase acitivities of traditional and industrial *doenjang*. The asterisks above the bars indicate significant difference, according to Levene’s test (*p* < 0.05).

**Figure 4 foods-10-01377-f004:**
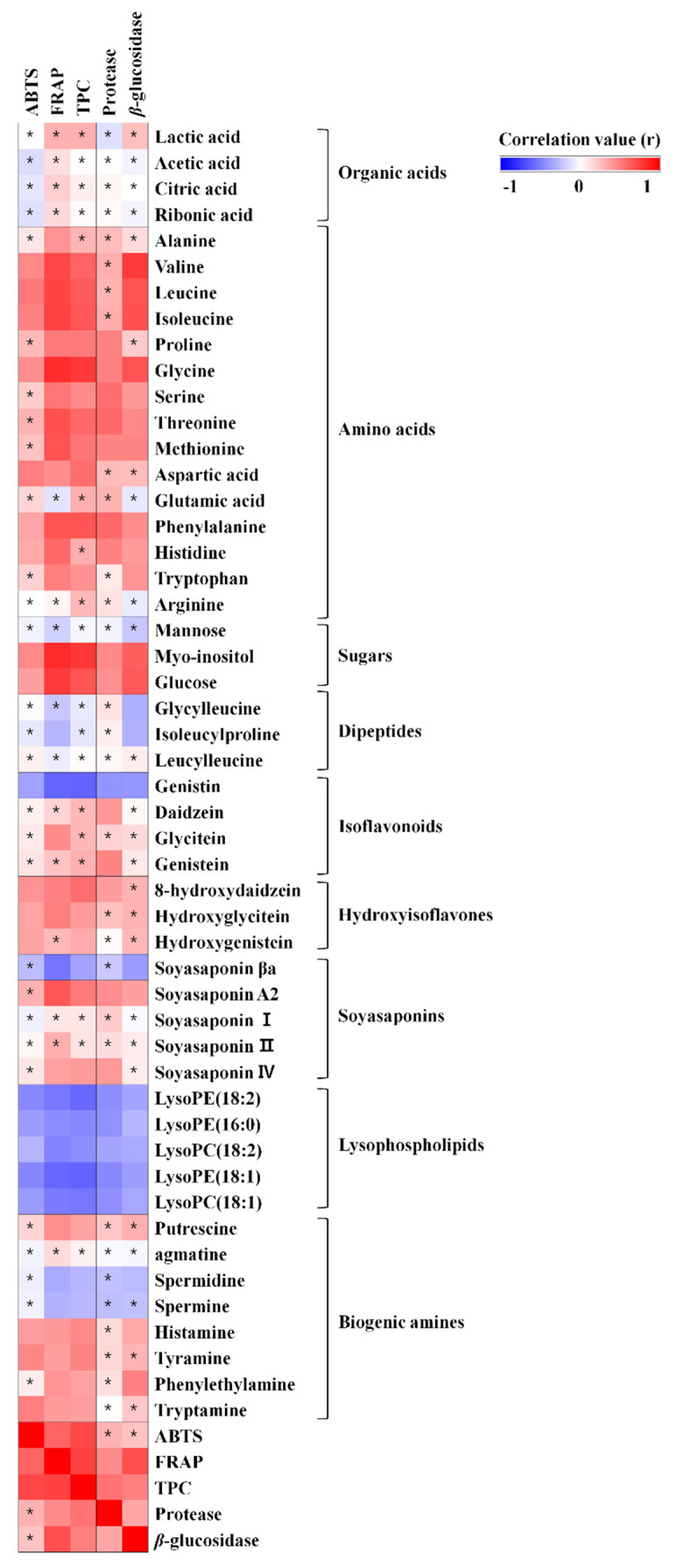
Correlation map between biochemical phenotypes and metabolites. Each square indicates Pearson’s correlation coefficient values (r). Red and blue represent positive (0 < r < 1) and negative (−1 < r < 0) correlation, respectively. Asterisks indicate significant difference (*p* < 0.05).

## Data Availability

The data presented in this study are available on request from the corresponding author.

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
