# Peer review of "Metabolomic-Based Comparison of Traditional and Industrial Doenjang Samples with Antioxidative Activities"

_foods, 2021, doi:10.3390/foods10061377_

Round 1

Reviewer 1 Report

The paper gives a detailed overview, i.e. analysis of the nutritional components of fermented soybean paste, i.e. Doenjang products. The introduction is clearly designed and gives a detailed insight into the problem and scientific interest of the research.
The materials and methods are clearly described, also the table in the supplementary files provides a detailed description of each sample, which is important to identify any differences in ingredients, preparation method and the like.
Appropriate multivariate methods of statistical analysis were used to possibly determine differences between traditionally and industrially processed doenjang, as well as correlations between metabolome and biochemical phenotypes.
The results are clearly presented and attached to the figures, I only suggest either redesigning Figure 2 or choosing a different way of presentation so that the results are visible. The discussion follows the results and the authors cite relevant research to explain the composition of various doenyang products.

Finally, I suggest adding a sentence that the authors would use to explain or highlight the main reason for the difference between traditionally prepared and industrially processed products.

Reviewer 2 Report

The manuscript is well designed, and the results reported are discussed in depth and relevant for the food science.

The major issue is related to the conclusion section, which seem to more a summary than reporting the real conclusion of the manuscript. 

Minor concerns:

Line 16: non-DDMP soyasaponins, could you describe the acronym the first time?

Lin 121: The MS data were collected in the range of 100–1500 m/z. Were you expecting metabolites above 1000 Daltons?

Line 121: The MS data were collected in the range of 100–1500 m/z (under negative and positive ion modes)

Line 128: I would add “independently” to explain that GC and LC data were independently treated.

Line 137: The discriminant metabolites were selected based on variable importance in the projection (VIP) value > 0.7. How was this threshold defined?

Figure 1: make it sharper. LC positive or negative ionization mode?

Figure 1B: Q2 is 0.08. Is it a typo for 0.8?

Figure 1C: Q2 is very low 0.318 for both PCA and PLS-DA models

Table S3: some mass errors are missing, i.e. L-Phenylalanine

-5 metabolites elute at the same rt, which is 0.78. It seems that they are not retained by the chromatographic column.

-The MS/MS fragments were acquired in pos or negative mode? I would suggest specifying it. Normally PE and PC are detected in positive ion mode. PC are indeed confirmed in positive mode with some characteristic fragments, such as those deriving from choline, including m/z 184, m/z 104.

-I would suggest adding bot the VIP values and the p-values for the LC significant metabolites

Table S2 I would suggest adding the p-values since both VIP and p-values were used to select these metabolites.
